# Multiplex Protein Biomarker Profiling in Patients with Familial Hypercholesterolemia

**DOI:** 10.3390/genes12101599

**Published:** 2021-10-12

**Authors:** Dana Dlouha, Milan Blaha, Eva Rohlova, Jaroslav A. Hubacek, Vera Lanska, Jakub Visek, Vladimir Blaha

**Affiliations:** 1Center for Experimental Medicine, Institute for Clinical and Experimental Medicine, Videnska 1958/9, 140 21 Prague, Czech Republic; eva.rohlova@ibt.cas.cz (E.R.); jahb@ikem.cz (J.A.H.); 24th Department of Internal Medicine—Hematology, Faculty of Medicine in Hradec Králové, University Hospital Hradec Králové and Charles University, 500 05 Hradec Králové, Czech Republic; milan.blaha@fnhk.cz; 3Laboratory of Gene Expression, Institute of Biotechnology CAS, BIOCEV, 252 50 Vestec, Czech Republic; 4Department of Anthropology and Human Genetics, Faculty of Science, Charles University, 120 00 Prague, Czech Republic; 5First Faculty of Medicine, Charles University, 128 00 Prague, Czech Republic; 6Statistical Unit, Institute for Clinical and Experimental Medicine, 140 21 Prague, Czech Republic; vela@ikem.cz; 73rd Department of Internal Medicine—Metabolism and Gerontology, Faculty of Medicine in Hradec Králové, University Hospital Hradec Králové and Charles University, 500 05 Hradec Králové, Czech Republic; jakub.visek@fnhk.cz (J.V.); blaha@lfhk.cuni.cz (V.B.)

**Keywords:** biomarker, familial hypercholesterolemia, apheresis, statins, protein

## Abstract

Familial hypercholesterolemia (FH), is an autosomal dominant disorder caused by mutations in the *LDLR*, *APOB*, *PCSK9*, and *APOE* genes and is characterized by high plasma levels of total and low-density lipoprotein (LDL) cholesterol. Our study aimed to analyze the influences of two different therapies on a wide spectrum of plasma protein biomarkers of cardiovascular diseases. Plasma from FH patients under hypolipidemic therapy (*N* = 18; men = 8, age 55.4 ± 13.1 years) and patients under combined long-term LDL apheresis/hypolipidemic therapy (*N* = 14; men = 7; age 58.0 ± 13.6 years) were analyzed in our study. We measured a profile of 184 cardiovascular disease (CVD) associated proteins using a proximity extension assay (PEA). Hypolipidemic therapy significantly (all *p* < 0.01) influenced 10 plasma proteins (TM, DKK1, CCL3, CD4, PDGF subunit B, AGRP, IL18, THPO, and LOX1 decreased; ST2 increased). Under combined apheresis/hypolipidemic treatment, 18 plasma proteins (LDLR, PCSK9, MMP-3, GDF2, CTRC, SORT1, VEGFD, IL27, CCL24, and KIM1 decreased; OPN, COL1A1, KLK6, IL4RA, PLC, TNFR1, GLO1, and PTX3 increased) were significantly affected (all *p* < 0.006). Hypolipidemic treatment mainly affected biomarkers involved in vascular endothelial maintenance. Combined therapy influenced proteins that participate in cholesterol metabolism and inflammation.

## 1. Introduction

Familial hypercholesterolemia (FH) is an autosomal dominant disorder characterized by elevated plasma levels of low-density lipoprotein (LDL-C) cholesterol in the absence of hypertriglyceridemia. FH is caused by mutations that lead to reduced function of the LDL receptor, with the most common being mutations in the *LDLR* gene itself. Less commonly, the FH phenotype may be caused by mutations in other genes, specifically apolipoprotein B (*APOB*), which encodes the ligand of the LDL receptor, and *PCSK9*, which encodes the enzyme proprotein convertase subtilisin/kexin type 9, which is involved in regulating the degradation of the LDL receptor; very rarely, mutations also occur in other genes, e.g., *APOE* [1]. Untreated FH is associated with a markedly increased risk of premature cardiovascular disease (CVD) depending on the specific molecular defect, the level of LDL-C, and coexisting cardiovascular risk factors [2,3,4].

FH patients should always be actively treated to lower plasma levels of LDL-C by diet and changes in lifestyle but also require pharmacological therapy for effective LDL-C control [5].

Statins are the first-choice drugs for FH patients and are used extensively in the treatment of dyslipidemia and in the long-term prevention of coronary artery disease (CAD) and stroke. Many FH patients cannot achieve adequate control of LDL-C levels with high-intensity statin therapy, and a cholesterol absorption inhibitor (ezetimibe) and PCSK9 inhibitor are the next-line classes of drugs [6].

Lipoprotein apheresis should be considered a therapeutic option for patients with severe hypercholesterolemia who have persistently elevated LDL-C levels despite attempts at drug therapy [7]. It is an extracorporeal elimination technique that removes LDL particles and other pathogenic lipoproteins, such as lipoprotein(a) or triglyceride-rich lipoproteins, from the circulation. The main indications for lipoprotein apheresis are, firstly, the diagnosis of homozygous FH, secondly, heterozygous FH that is refractory to the standard care and intolerant to routine care, and, thirdly, patients with lipoprotein(a) and increased resistance to pharmacotherapy [8]. Lipoprotein apheresis is also a potent therapy that impacts inflammation and related mediators [9].

The aim of our study was to evaluate the cholesterol-independent effects of different types of hypolipidemic intervention on plasma levels of 184 CVD—related proteins in FH patients.

## 2. Materials and Methods

### 2.1. Subjects

In our study were separately measured two groups of patients (Table 1): FH patients under lipid lowering drugs therapy (LLD^+^), without apheresis treatment (AF^−^) (LLD^+^/AF^−^; *N* = 18; men = 8, aged 55.4 ± 13.1 years) and patients under combined long-term LDL apheresis/LLD (LLD^+^/AF^+^; *N* = 14; men = 7; aged 58.0 ± 13.6 years). Both groups were analyzed independently.

LLD^+^/AF^−^ patients were treated daily with atorvastatin (10–40 mg), rosuvastatin (10–40 mg), simvastatin (20 mg), and ezetimibe (10 mg).

The characteristics and clinical phenotype of LLD^+^/AF^+^ patients have been previously described [10]. DNA-based evidence of a mutation in the *LDLR* gene was the criterion for the diagnosis of homozygous FH. None of the patients had a mutation in the *APOB* gene. The patients had been regularly treated with LDL apheresis (immunoadsorption) or rheohemapheresis (cascade filtration) for an average of 12.3 ± 6.6 years. All patients were treated daily with statins (rosuvastatin 40 mg or atorvastatin 80 mg), 1 patient was treated in combination with fenofibrate (160 mg), 2 patients were treated in combination with bile acid-binding resins (6 mg), all patients were treated in combination with ezetimibe (10 mg), and at the time of sampling 3 patients were treated with PCSK9 inhibitors.

The protocol was carried out according to the Declaration of Helsinki. All examined individuals were Caucasians, and all patients signed informed consent forms, which, together with the protocol of the study, were approved by the institute’s ethics committee. The basic characteristics of the patients are summarized in Table 1.

### 2.2. LDL Apheresis

Plasma separation was performed using a Cobe-Spectra or Optia continuous centrifugal separator (Terumo, Likewood, CO, USA) in 9 patients. An adsorption-desorption automatic device (Adasorb, Medicap, Germany) controlled repeated fillings and washings of Lipopak adsorbers (Pocard, Moscow, Russia). In 2 patients, Lipocollect adsorbers (Medicollect, Germany) were used. Briefly, patients’ blood was taken from a peripheral venous access to the blood cell separator Cobe-Spectra or Optia (Terumo, Likewood, CO, USA) that, acting as centrifuge, separates plasma and cellular components of the blood. In the immunoadsorption technique, plasma was pumped through affinity columns Lipopak (Pocard, Moscow, Russia), containing antibodies against the main lipoprotein of LDL-cholesterol—apolipoprotein B100 [11].

### 2.3. Rheohemapheresis

Three patients simultaneously received long-term therapy due to hypercholesterolaemia and increased levels of fibrinogen. Rheohaemapheresis therapy was used according to Borberg et al. with our own modification [12]. In rheopheresis, plasma is pumped through a filter that separates the lipoproteins and other large molecules. On the basis of the hypothesis that the adsorption column removed atherosclerosis-related proteins other than lipoprotein-binding proteins or positively charged proteins, proteomic analysis of the waste fluid was performed. The 48 kinds of proteins in the waste fluid of LDL adsorption columns, including coagulation factors, thrombogenic factors, complement factors, inflammatory factors, and adhesion molecules, were identified [13]. Purified plasma is mixed with blood cells separated earlier, and returned back to the patient via another peripheral vein. The adsorption is fully automated; the plasma flow through the adsorption columns is directed by a secondary device, ADA or Adasorb (Medicap, Ulrichstein, Germany). To obtain plasma, we used continuous separators (Cobe Spectra or Spectra Optia, Terumo BCT, Lakewood, Co, USA) and Evaflux 4A filters (Kawasumi, Tokyo, Japan) to wash the obtained plasma were used. The flow through the filter was controlled, using the CF100 automatic machine (Infomed, Geneva, Switzerland). Anticoagulation was performed using a combination of heparin and ACD-A (Baxter, Munich, Germany). Then, 1–1.5 of circulating plasma volume, calculated by the computer, of blood cell separator was washed. The procedures were performed from the peripheral vein in the elbow pit or in the forearm.

### 2.4. Plasma Samples and Blood Analysis

Venous blood (10 mL) was collected in EDTA-containing tubes and centrifuged at 1500× *g* for 15 min at room temperature. Plasma samples were processed within 30 min of blood collection, aliquoted into RNAse/DNAse-free tubes and stored for 6–12 months at −80 °C before proteomic analysis. Samples from LLD^+^/AF^−^ patients were collected prior to the start of pharmacotherapy and then 1 month later. Samples from LLD^+^/AF^+^ patients were collected before and after apheresis during periodic apheresis treatment.

Lipid parameters included total cholesterol, HDL-C, triglycerides, and LDL-C levels were direct LDL-C measurement was performed. Non-HDL-C was calculated by subtracting HDL-C to total cholesterol. Lipid parameters were measured using an enzymatic colorimetric method, with intra- and inter-assay coefficients of variation of <2.8% and <3.9%, respectively. These parameters were measured using an automated autoanalyzer (Cobas 8000, Roche Diagnostics, Mannheim, Germany). Quality control was performed according to hospital standards [14]. 

LDL-C values for LLD^+^/AF^+^ patients are not possible to correct for LLD use. The LDL apheresis is the limiting reason due to the individually different rebound of lipoproteins after LDL-apheresis [15,16]. Acute decreases in LDL-cholesterol after each procedure range from 60% to 80%, depending upon the volume of blood or plasma treated. The subsequent rebound in plasma cholesterol is fastest in normal subjects and slowest in FH homozygotes, with heterozygotes intermediate.

### 2.5. Proteomic Analysis

Analyses were performed using a high-throughput technique using Olink Proseek^®^ Multiplex CVD II and CVD III panels (https://www.olink.com/products/ (accessed on 25 July 2018)) at the TATAA Biocenter (Odinsgatan 28, SE-411 03 Göteborg, Sweden). Each panel included 92 preselected CVD biomarkers. These panels contain known human cardiovascular and inflammatory markers, as well as some exploratory human proteins with great potential as new CVD markers, which were carefully selected in collaboration with leading experts in the field. Each analyte in the panel has been assessed in terms of sample material, specificity, precision, sensitivity, dynamic range, matrix effects, and interference. The biomarker pages (https://www.olink.com/content/uploads/2019/12/Olink-CVD-II-Validation-Data-v2.1.pdf (accessed on 20 March 2020) and https://www.olink.com/content/uploads/2019/12/Olink-CVD-III-Validation-Data-v2.1.pdf (accessed on 20 March 2020)) include calibrator curves that show the performance of each assay with the estimated sensitivity and dynamic range parameters indicated. These curves are generated during the assay validation process using recombinant antigens, with data presented as normalized protein expression (NPX) values plotted against protein concentration (in pg/mL).

The kit used a proximity extension assay (PEA) technology [17]. Briefly, plasma samples (1 µL) were incubated with 92 oligonucleotide labeled antibody probe pairs that bind to their respective target present in the sample [18]. A PCR reporter sequence was formed by a proximity dependent DNA polymerization event and was subsequently detected and quantified using high-throughput real-time PCR (BioMark™ HD System, Fluidigm Corporation). Data analysis was performed by employing a pre-processing normalization procedure. For each data point, delta Cq (dCq) values were obtained by subtracting the value for the extension control, thus normalizing each sample for technical variation within one run. Normalization between runs was then performed by subtraction of the interplate control for each assay. In the final step of the pre-processing procedure, the values were set relative to a correction factor determined by Olink. The generated normalized protein expression (NPX) unit is on a log2 scale where a larger number represents a higher protein level in the sample. Linearization of the protein expression data (linear ddCq) was performed by the mathematical operation 2^NPX^. Multiplex CVD II^96×96^ and CVD III^96×96^ showed mean intra-assay coefficients of variation (CVs) of 9.1% and 8.1% and interassay coefficients of variation of 11.7% and 11.4%, respectively. Nine proteins (AZU1, SPON1, CCL22, SLAMF7, STK4, ITGB1BP2, BNP, PAPPA, and CA5A) with a call rate <75% (i.e., less than 75% of the individuals had a valid measurement of that protein) were removed from further analysis (Appendix A). Using the CVD II and CVD III PEA panels, 92 low-abundance plasma proteins and 92 high-abundance proteins relevant to CVD, respectively, were quantitated in plasma samples. The protein markers included in the CVD II and CVD III assays are available at https://www.olink.com/content/uploads/2017/07/1024-v1.3-CVD-II-Panel-Content_final.pdf (accessed on 25 July 2018) and https://www.olink.com/content/uploads/2017/09/1023-v1.2.1-CVD-III-Panel-Content_final.pdf (accessed on 25 July 2018).

### 2.6. Statistical Analyses

All data were converted to a log2 scale before statistical analyses. GenEx SW (Multid Analysis AB, Göteborg, Sweden) and JMP statistical software (2012 SAS Institute, Inc., Cary, NC, USA) were used for statistical analysis. T-test (mean values with SDs) was used for intergroup comparison and Wilcoxon matched-paired signed rank test was used for intragroup comparison and then Bonferroni correction was applied on significance levels. The significance level was set to *p* < 0.01. Spearman’s ρ was used for correlation analyses.

## 3. Results

### 3.1. Lipid Lowering Drugs Therapy—Only

Firstly, we compared plasma level of biomarkers at a zero point of treatment and 1 month after using drugs in LLD^+^/AF^−^ group. We found 10 proteins (TM, DKK1, CCL3, CD4, PDGF subunit B, AGRP, IL18, THPO, and LOX1; all *p* < 0.01) significantly decreased 1-month after therapy, only ST2 was increased (Figure 1A and Appendix A). Of 184 investigated protein biomarkers, 19 demonstrated association with at least 1 of the lipid fractions (Table 2a). Of these, the most significant relationship was found between the soluble LDLR and TAG levels before and after treatment (*p* < 0.01, resp. *p* < 0.0001). Further, IGFBP-2 related to both TAG and HDL-C in a consistent and biologically expected manner, that is, lower TAG (*p* < 0.0001) and higher HDL-C (*p* < 0.005). PRSS27 and CCL3 were associated with TC and LDL-C fractions after treatment (all *p* < 0.005).

### 3.2. Combined Long-Term LDL Apheresis/LLD Therapy

In LLD^+^/AF^+^ patients, concentration of 18 protein biomarkers was affected. LDLR, PCSK9, MMP-3, GDF-2, CTRC, SORT1, VEGFRD, IL-27, CCL24, and KIM1 was reduced (all *p* < 0.006) and concentration of OPN, COL1A1, KLK6, IL4RA, PLC, TNFR1, PTX3, and GLO1, was increased (all *p* < 0.005) after a single standard therapy. The most significant decline was detected in circulating soluble LDLR, PCSK9, and MMP-3 (all *p* < 0.0002; Figure 1B and Appendix A). The opposite trend was observed in COL1A1 and OPN (both *p* < 0.0003). After Bonferroni correction remain significantly deregulated only LDLR, OPN, PCSK9, MMP-3, and COL1A1 (all *p* < 0.0003) in LLD+/AF+ patients. We also identified differences in LDLR between *LDLR* heterozygotes vs. homozygotes (Appendix A).

In LLD^+^/AF^+^ group, 32 protein biomarkers were associated with at least 1 of the plasma lipid fractions (Table 2B). Among these associations we found REN, AXL, MB, IGFBP-7, CDH5, and CCL15 to be inversely associated with both, TC and LDL-C before apheresis therapy. Of associated proteins, REN, KIM1, and GDF-15 significantly correlated with plasma lipid levels before and after treatment (all *p* < 0.01). The most significant relationship was found between ACE2 and TAG after apheresis therapy (*p* < 0.0001).

## 4. Discussion

The aim of our study was to track the influence of different therapies for FH on typical circulating CVD-related protein biomarkers. The analysis showed a predicted decline of plasma lipid concentration (Appendix A) in both groups after treatment. In recent years, experimental and clinical evidence have suggested that the beneficial effects of statins, ezetimibe or PCSK9 inhibitors against atherosclerosis might be due to cholesterol-independent effects (improving endothelial function, stabilizing atherosclerotic plaques, attenuating vascular and myocardial remodeling, and inhibiting vascular inflammation, oxidation, and thrombosis) [19].

Our large proteomic study detected variability in protein regulation in response to diverse therapies. The majority of proteins that changed in LLD^+^/AF^−^ patients are involved in vascular endothelial maintenance. Combined LLD^+^/AF^+^ treatment predominantly influenced proteins that participate in cholesterol metabolism and inflammation (Appendix A).

### 4.1. Lipid Metabolism

A decline in only one protein involved in lipid metabolism, the primary receptor for ox-LDL in endothelial cells, LOX-1, was found in LLD^+^/AF^−^. Matarrazo et al. showed that long-term exposure to different statins results in chronic inhibition of cholesterol biosynthesis and leads to a marked reduction in LOX-1 in lipid rafts and a consequent reduction in ox-LDL binding and uptake [20]. Contrary, in LLD^+^/AF^+^ was found significant decline in majority of proatherogenic markers, such as LDLR, PCSK9, MMP-3, VEGF-D, and SORT1, which are implicated in LDL cholesterol metabolism, VLDL and PCSK9 secretion and the development of atherosclerotic lesions [21,22]. Finally, the same patterns were observed in GDF-2, which is implicated in the regulation of the hepatic reticuloendothelial system, glucose homeostasis, and inhibition of angiogenesis, and a serine protease regulator CTRC, that is involved in lipoprotein metabolism [23].

### 4.2. Inflammation

Inflammation has been postulated to play an important role in atherogenesis. A wide range of cytokines have been investigated both as risk markers and as possible risk factors for atherosclerosis development.

In LLD^+^/AF^−^, we found downregulated CCL3 and IL-18, both contributing to atherosclerosis, e.g., by inducing the recruitment of monocytes from the bloodstream to the vascular endothelium, causing the formation of foam cells, and promoting the retention and differentiation of leukocytes within the vascular endothelium or by proinflammatory effect [24,25]. Elevated levels of ST2 suggest the adverse cardiac remodeling and tissue fibrosis, which occurs in response to myocardial infarction, acute coronary syndrome, or worsening heart failure [26].

Of the cytokines that were downregulated in response to apheresis treatment, we found proinflammatory IL-27 and profibrotic CCL24 in LLD^+^/AF^+^ patients. Circulating IL-27 is associated with oxLDL in plasma and CVD severity [27]. CCL24 promotes immune cell trafficking and activation, as well as profibrotic activities through the C-C chemokine receptor type 3 [28]. Apheresis treatment also reduced plasma KIM1, a transmembrane protein and marker of renal tubular injury, which has been previously mentioned as a cardiovascular risk factor [29]. Further, IL-4RA inhibits adipogenesis and promotes lipolysis by enhancing the activity of hormone-sensitive lipase [30]. Increased levels of the anti-inflammatory cytokine IL-4RA suggest the positive influence of single apheresis treatment.

In LLD^+^/AF^+^, we also detected the elevation of proatherogenic markers (OPN, TNFR1, PTX3, and KLK6). Increased OPN plasma levels were previously reported to be associated with the presence and extent of coronary artery disease, are independent predictors of future adverse cardiac events in patients with chronic stable angina [31]. Proinflammatory TNFR1 contributes to the pathogenesis of atherosclerosis by enhancing arterial wall chemokine and adhesion molecule expression [32]. Increased plasma PTX3 levels were reported in patients with carotid stenosis and acute coronary syndrome, and PTX3 is also a candidate biomarker for plaque vulnerability [33]. A serine protease KLK6, is increased in response to neuroinflammation and regulates immune cell survival [34].

### 4.3. Vascular Endothelium

Recently, numerous studies have shown that the cholesterol-independent vascular effects of statins appear to involve directly restoring or improving endothelial function by increasing NO production, promoting re-endothelialization after arterial injury, and inhibiting inflammatory responses within the vessel wall that are thought to contribute to atherosclerosis [35]. Plasma levels of biomarkers involved in the regulation of endothelial function were detected downregulated in LLD^+^/AF^−^. Of them, a decline in TM was observed in several pathological conditions in which the endothelium is likely to be damaged, such as atherosclerosis, hypertension, coronary artery disease, or ischemic stroke [36]. DKK-1, that contributes to the early stages of atherogenesis, was the second most downregulated biomarker after 1 month of drugs only therapy. A significant positive correlation between plasma DKK-1 and carotid artery intima-media thickness, was reported [37]. PDGF subunit B, that is known to contribute to macrophage, platelet, smooth muscle cell, and fibroblast migration and proliferation within blood vessels, as well as in atherosclerotic lesions were regulated in response to statin therapy. Statins may attenuate PDGF-induced mitogenesis in plaques and limit deleterious mitogenic responses that lead to the migration and activation of smooth muscle cells and macrophages [38].

### 4.4. Thromboembolism

The potential role of statins in reducing the incidence of venous thromboembolism has been reported [39]. We detected a decline in plasma THPO and AGRP levels in LLD^+^/AF^−^ in response to hypolipidemic therapy. THPO is the major cytokine that regulates platelet production, controls the proliferation and differentiation of megakaryocyte progenitor cells and is essential for the maintenance of thrombopoiesis. Elevated plasma THPO levels have been reported in patients with acute coronary syndromes [40]. AGRP is downregulated through the reduction in KLF4 expression, which is a potent activator of AGRP [41].

In LLD^+^/AF^+^, we found increased plasma levels of COL1A1, PLC, and GLO1. COL1A1 is the major component of fibrillar collagen found in most connective tissues, including cartilage, and was identified as a potential biomarker for heart failure progression [42]. PLC, a large heparan sulfate proteoglycan, a major component of the vessel wall, has been linked to lipid retention in the vasculature [43]. GLO1 is the main opponent of the degradation of the reactive metabolite methylglyoxal, and plays an important role in the progression of atherosclerosis and plaque rupture [44].

### 4.5. Lipid-Protein Associations

We identified an overlap of proteins associated with plasma lipids (Table 2). We found a positive correlation between IGFBP-2 and HDL before, and an inverse association between IGFBP-2, IGFBP-1 and TAG levels after hypolipidemic treatment in LLD+/AF^−^ patients (Table 2A). Both proteins participating in adipogenesis and lipogenesis pathways and are related to fat mass and insulin sensitivity [45].

Further, eight protein biomarkers were identified to be associated with TC and LDL-C. The inverse correlation between predominantly pro-inflammatory proteins and TC and LDL-C before apheresis treatment and positive association after hypolipidemic treatment in LLD^+^/AF^−^ patients reflect the effect of both therapies.

The pro-inflammatory GDF-15, a macrophage inhibitory cytokine, and KIM1 were found associated with TAG in LLD^+^/AF^+^ patients. The pro-inflammatory function of these proteins could relate to coronary plaque development and the occurrence of events in FH patients.

The concept of statin pleiotropy has provided a window of opportunity to test and target other non-lipid-lowering signaling pathways that may affect cardiovascular disease. We have shown, that drugs only treatment mainly affected biomarkers that participate in muscle injury, the regulation of coagulation and fibrinolysis. Our results are in an agreement with known statin pleiotropic effects that modulate coagulation and inflammation. Moreover, our study is the first, that reported a direct relationship between the reduction in DKK1 in plasma as a response to statin treatment. The majority of influenced proteins are associated with inflammatory pathways and cholesterol metabolism, both key for angiogenesis, myocardial ischemia, and atherosclerosis progression. On the other hand, increased plasma levels of some pro-atherogenic cytokines and proteases in response to single apheresis treatment suggest, also, some potentially deleterious effects of combined LDL apheresis/hypolipidemic therapy.

The limitations of this single-center prospective observational study include the relatively small number of patients, especially in the lipoprotein apheresis treatment group, and we did not incorporate a control treatment arm in this study. The small number of participants in the present study may affect the accuracy of our results. Thus, our findings need to be verified by multicenter prospective studies. In addition, we cannot exclude the influence of concomitant treatment besides lipoprotein apheresis (statins, ezetimibe, PCSK9 inhibitors) on the measured parameters. Furthermore, since the lipoprotein apheresis treatment technique is carried out only in a few big medical centers, many FH patients are unable to receive apheresis treatment, resulting in a particular bias in patient selection.

## 5. Conclusions

Our large proteomic study detected variability in protein regulation in response to diverse therapies. The majority of proteins that changed in LLD^+^/AF^−^ patients are involved in vascular endothelial maintenance. Combined LLD^+^/AF^+^ treatment predominantly influenced proteins that participate in cholesterol metabolism and inflammation.

## Figures and Tables

**Figure 1 genes-12-01599-f001:**
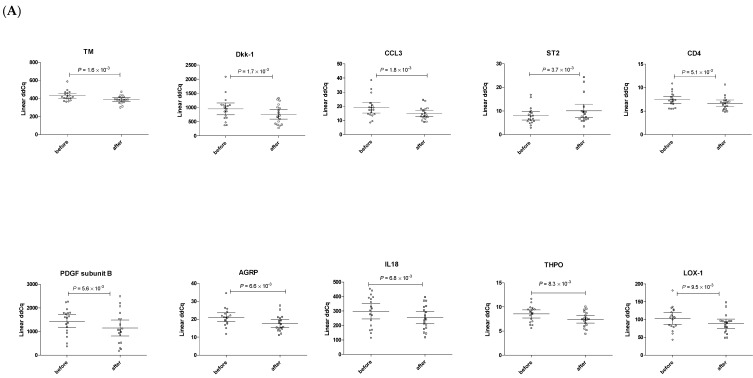
Distribution of plasma levels of top-deregulated protein biomarkers in response to therapies. (**A**)/LLD^+^/AF^−^ patients; (**B**)/LLD^+^/AF^+^ patients. Scatter plot, group means (before and after treatment) are indicated by horizontal bars, error bars indicate 95% CI; the significance level was set to *p* < 0.01.

**Table 1 genes-12-01599-t001:** Basic characteristics of patients.

	LLD^+^/AF^−^	LLD^+^/AF^+^
Male/Female (N)	8/10	7/7
Age (years)	55.4 ± 13.1	58.0 ± 13.6
BMI (kg/m^2^)	26.5 ± 4.1	26.8 ± 3.1
Current smokers and ex-smokers/non-smokers	7/11	4/10
Hypertension	5	5
Diabetes Mellitus	0	2
LDLR mutations: (N) homozygotes/heterozygotes	0/18	6/8
Duration of apheresis treatment (years)	-	12.3 ± 6.5
Hypolipidemic treatment	1 month	>13 years
ACE-i/ARB	4	3
Betablockers	4	6
Antithrombotic drugs	3	14
PCSK9 inhibitors	0	3
Statins	8 (Atoravastatin 10–40 mg) 7 (Rosuvastatin 10–40 mg) 2 (Atoravastatin/Rosuvastatin 40 mg + Ezetimibe 10 mg) 1 (Simvastatin 20 mg)	14 (all Rosuvastatin 40 mg or Atorvastatin 80 mg + Ezetimibe 10 mg)

Data are expressed as mean ± SD or factor proportion. SD—standard deviation, BMI—body mass index.

**Table 2 genes-12-01599-t002:** Nonparametric Spearman’s correlation of overall plasma lipids and protein biomarkers: (**a**) LLD^+^/AF^−^ group; (**b**) LLD^+^/AF^+^ group.

**(a)/LLD^+^/AF^−^**	**Before**			**After**	
**Protein**	**Lipids**	**Spearman ρ**	** *p* **	**Spearman ρ**	** *p* **
PTX3	TC	−0.5356	0.0084	−0.1536	0.5302
PRSS27	TC	0.2905	0.1787	0.7041	0.0008
PRSS27	LDL-C	0.2881	0.1825	0.6570	0.0022
MB	TC	−0.2223	0.3079	−0.6319	0.0037
CCL3	TC	0.1423	0.5172	0.6222	0.0044
CCL3	LDL-C	0.1769	0.4193	0.5925	0.0075
GDF-2	TC	0.3262	0.1288	0.5801	0.0092
MARCO	LDL-C	0.0919	0.6766	0.7068	0.0007
CD40-L	LDL-C	0.6123	0.0019	0.1434	0.5582
PON3	HDL-C	0.5871	0.0032	0.3696	0.1193
IGFBP-2	HDL-C	0.5660	0.0049	0.5470	0.0154
PSP-D	HDL-C	0.5368	0.0083	0.5312	0.0193
GLO1	HDL-C	−0.2990	0.1658	−0.6839	0.0012
IL-1ra	HDL-C	−0.4359	0.0376	−0.6471	0.0027
NT-proBNP	HDL-C	0.4398	0.0357	0.6383	0.0033
GH	HDL-C	0.2100	0.3361	0.5909	0.0077
LDL receptor	TAG	0.5649	0.0050	0.8228	<.0001
IGFBP-2	TAG	−0.5156	0.0118	−0.7895	<.0001
LPL	TAG	−0.4563	0.0287	−0.7474	0.0002
IGFBP-1	TAG	−0.3326	0.1210	−0.6105	0.0055
ACE2	TAG	0.3420	0.1102	0.5965	0.0070
t-PA	TAG	0.3706	0.0817	0.5825	0.0089
**(b)/LLD^+^/AF^+^**	**Before**			**After**	
**Protein**	**Lipids**	**Spearman ρ**	** *p* **	**Spearman ρ**	** *p* **
REN	TC	−0.7357	0.0018	−0.5380	0.0386
REN	LDL-C	−0.6500	0.0087	−0.6810	0.0052
AXL	TC	−0.6964	0.0039	−0.4383	0.1022
AXL	LDL-C	−0.6429	0.0097	−0.4705	0.0767
MB	TC	−0.6929	0.0042	0.0250	0.9295
MB	LDL-C	−0.7571	0.0011	0.1233	0.6615
IGFBP-7	TC	−0.6571	0.0078	0.0697	0.8050
IGFBP-7	LDL-C	−0.7036	0.0034	0.1698	0.5452
CDH5	TC	−0.6464	0.0092	−0.2574	0.3544
CDH5	LDL-C	−0.6857	0.0048	−0.1573	0.5756
CCL15	TC	−0.6464	0.0092	0.1787	0.5239
CCL15	LDL-C	−0.7286	0.0021	−0.2055	0.4624
PGF	TC	−0.7250	0.0022	−0.1823	0.5155
CXCL1	TC	0.6893	0.0045	0.2341	0.4010
MCP-1	LDL-C	−0.6964	0.0039	−0.3342	0.2234
IL-18BP	LDL-C	−0.6702	0.0063	−0.1144	0.6848
AP-N	LDL-C	−0.6679	0.0065	−0.5451	0.0356
TLT-2	LDL-C	−0.6464	0.0092	−0.0465	0.8694
uPA	LDL-C	−0.6464	0.0092	0.1555	0.5800
KIM1	TAG	0.7750	0.0007	0.7006	0.0036
RARRES2	TAG	0.7607	0.0010	0.4093	0.1298
FGF-21	TAG	0.7607	0.0010	0.5541	0.0321
GDF-15	TAG	0.6786	0.0054	0.7310	0.0020
t-PA	TAG	0.6429	0.0097	0.2788	0.3143
IL-4RA	TAG	−0.6429	0.0097	−0.2522	0.3644
IL16	TC	0.0858	0.7611	0.7775	0.0006
PIgR	TC	0.0071	0.9798	0.7685	0.0008
vWF	TC	0.3571	0.1913	−0.7328	0.0019
CEACAM8	TC	−0.2000	0.4748	0.6816	0.0051
ADAM-TS13	TC	−0.0214	0.9396	−0.6780	0.0055
LOX-1	TC	−0.0769	0.7854	0.6488	0.0089
PON3	HDL-C	0.5571	0.0310	0.6500	0.0087
ACE2	TAG	0.5143	0.0498	0.8472	<.0001
Notch 3	TAG	−0.5607	0.0297	−0.7489	0.0013
PRSS8	TAG	0.2750	0.3212	0.7435	0.0015
TR	TAG	−0.1464	0.6025	−0.7024	0.0035
FGF-23	TAG	−0.0250	0.9295	0.6720	0.0061
IL-1RT2	TAG	0.2643	0.3412	0.6506	0.0086

## Data Availability

All data that support the findings of this study are available from the corresponding author upon reasonable request.

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
