# Peer review of "Multiplex Protein Biomarker Profiling in Patients with Familial Hypercholesterolemia"

_genes, 2021, doi:10.3390/genes12101599_

Round 1

Reviewer 1 Report

Dear Editors/authors, Thank you for the opportunity to review the manuscript, " Multiplex protein biomarker profiling in patients with familial hypercholesterolemia" by Dana Dlouha and colleagues.   The present study focuses on patients with familial hypercholesterolemia (FH), a genetic disorder which leads to increase of LDL-cholesterol plasma levels and to subsequent increased risk for atherosclerotic cardiovascular diseases. Statins (usually high dose) are the first-choice drugs for FH patients and are usually combined with add-on lipid lowering therapies (ezetimibe and PCSK9 inhibitors). In extreme cases, e.g patients with homozygous-FH, intolerant or non-responders to highest tolerated dose of statins, LDL apheresis is in indicated. This technic consists in an extracorporeal elimination of circulating LDL particles. The authors study here circulating protein biomarkers (184 markers Olink technology) in: i) FH-patients before and after the initiation of lipid-lowering treatments (n=18), ii) FH-patients before and after the initiation of apheresis treatment (n=14).   The manuscript is straight forward and well written. While mainly descriptive it presents valuable data for the treating clinician as well as for the research community. I have minor comments and suggestions which in my view would provide additional information.    Minor comments:   i) In the methods section, the authors refer to a previous paper [REF 10, Dlouha et al. 2017] in which the clinical data of the LLD+/AF+ are presented. I would be interested in seeing the calculated LDL values for these patients (corrected for LLD use). ii) I recommend to add a supplemental table with FH causal mutations as seeing LDLR circulating in all these patients is intriguing me.  iii) Comparing circulating biomarkers of the 3 LLD+/AF+-PCSK9 treated patients with LLD+/AF+ might bring useful supplemental data. iv) My main concerns relate to the presentation of the O-link data. I suggest to add in a supplemental table with : i) all circulating proteins measured, with the main biological processes they are involved in (e.g: angiogenesis etc. see https://www.olink.com/products/target/cvd-ii-panel/ ), ii) their molecular weight. The latest will give information on the retention capacity of apheresis as mentioned in the manuscript (page 3 line 106). v) I have a methodological concern regarding the Bonferroni corrections. As mentioned in the abstract 184 CVD associated proteins were measured. Could the authors explain the significant threshold they used for figure 1A and B.  vi) Figure 1A and B are quite difficult to read with a lot of information. If the authors think it is relevant I would suggest to scaled their data (normalized by sex, age, BMI etc) and present it as forest plot. vii) I would appreciate to read the limitations of the present study (number of patients, T2Diabetes). I am also quite worried about the fact that FH patients above 50 years of age (LLD+AF-) just start lipid lowering drugs at this age. I would like to see this observation discussed as well.        Typos   i) Gene names should be written in italic letters. ii) STAP1 should be removed from the FH causative genes list [see ‘Introduction’ section] (Loaiza et al. 2020, PMID: 31996024) iii) Page 9 line 23: ‘patters’should read as patterns

Author Response

  1. Reviewer

Dear Editors/authors, Thank you for the opportunity to review the manuscript, " Multiplex protein biomarker profiling in patients with familial hypercholesterolemia" by Dana Dlouha and colleagues.   The present study focuses on patients with familial hypercholesterolemia (FH), a genetic disorder which leads to increase of LDL-cholesterol plasma levels and to subsequent increased risk for atherosclerotic cardiovascular diseases. Statins (usually high dose) are the first-choice drugs for FH patients and are usually combined with add-on lipid lowering therapies (ezetimibe and PCSK9 inhibitors). In extreme cases, e.g patients with homozygous-FH, intolerant or non-responders to highest tolerated dose of statins, LDL apheresis is in indicated. This technic consists in an extracorporeal elimination of circulating LDL particles.

The authors study here circulating protein biomarkers (184 markers Olink technology) in:

  1. i) FH-patients before and after the initiation of lipid-lowering treatments (n=18),
  2. ii) FH-patients before and after the initiation of apheresis treatment (n=14).  

The manuscript is straight forward and well written. While mainly descriptive it presents valuable data for the treating clinician as well as for the research community. I have minor comments and suggestions which in my view would provide additional information.    Minor comments:  

  1. i) In the methods section, the authors refer to a previous paper [REF 10, Dlouha et al. 2017] in which the clinical data of the LLD+/AF+ are presented. I would be interested in seeing the calculated LDL values for these patients (corrected for LLD use).

Dear reviewer, lipid parameters included total cholesterol, HDL-c, triglycerides, and LDL-c levels, where direct LDL-c measurement was performed. LDL values for LLD+/AF+ patients is not possible to correct for LLD use.  The LDL aphereses are the limiting reason due to the individually different rebound of lipoproteins after LDL-apheresis (Kroon AA, van't Hof MA, Demacker PN, Stalenhoef AF. The rebound of lipoproteins after LDL-apheresis. Kinetics and estimation of mean lipoprotein levels. Atherosclerosis. 2000 Oct;152(2):519-26. doi: 10.1016/s0021-9150(00)00371-3. PMID: 10998482). Acute decreases in LDL-cholesterol after each procedure range from 60 to 80%, depending upon the volume of blood or plasma treated. The subsequent rebound in plasma cholesterol is fastest in normal subjects and slowest in FH homozygotes, with heterozygotes intermediate.

LDL values of patients from both analyzed groups (LLD+/AF-, LLD+/AF+) are presented in Table S2 in Supplementary material. Predicted decline of plasma lipid concentration is also mentioned in Discussion (page 9, lines 5-6).

  1. ii) I recommend to add a supplemental table with FH causal mutations as seeing LDLR circulating in all these patients is intriguing me. 

Dear reviewer, thank you very much for your recommendation. The table was added to the Supplementary material and the text in Results was modified. We identified differences between heterozygotes vs homozygotes only within LLD+/AF+ patients. Similarly, as we reported in manuscript, correlation was detected between LDLR and TAG also only in LLD+/AF+ patients.

iii) Comparing circulating biomarkers of the 3 LLD+/AF+-PCSK9 treated patients with LLD+/AF+ might bring useful supplemental data.

Dear reviewer, thank you very much for this comment. Number of patients treated with PCSK9 inhibitors is too small for unbiased statistical analysis. Moreover, all PCSK9 treated patients represent group of LDLR homozygotes, thus, it seems not to be reasonable to compare them with heterozygotes.

  1. iv) My main concerns relate to the presentation of the O-link data. I suggest to add in a supplemental table with : i) all circulating proteins measured, with the main biological processes they are involved in (e.g: angiogenesis etc. see https://www.olink.com/products/target/cvd-ii-panel/ ), ii) their molecular weight. The latest will give information on the retention capacity of apheresis as mentioned in the manuscript (page 3 line 106).

i/ Dear reviewer, thank you very much for this comment. We briefly discuss within the text biological processes mainly involved in the atherosclerosis development focused on significantly deregulated proteins only. According to your recommendation we prepared additional Table S4 (Supplementary material) with molecular weight of deregulated proteins and their main biological processes based on the data available at https://www.olink.com.

ii/ On the basis of the hypothesis that the adsorption column removed atherosclerosis-related proteins other than lipoprotein-binding proteins or positively charged proteins, proteomic analysis of the waste fluid was performed. Several kinds of proteins in the waste fluid of LDL adsorption columns, including coagulation factors, thrombogenic factors, complement factors, inflammatory factors and adhesion molecules was identified (YUASA, Yumiko, Tsukasa OSAKI, Hisashi MAKINO, Noriyuki IWAMOTO, Ichiro KISHIMOTO, Makoto USAMI, Naoto MINAMINO a Mariko HARADA-SHIBA. Proteomic Analysis of Proteins Eliminated by Low-Density Lipoprotein Apheresis. Therapeutic apheresis and dialysis [online]. HOBOKEN: Blackwell Publishing, 2014, 18(1), 93-102 [cit. 2021-9-29]. ISSN 1744-9979. doi:10.1111/1744-9987.12056). We also discuss the potential retention capacity of apheresis/ rheopheresis in text of paragraph 2.3

  1. v) I have a methodological concern regarding the Bonferroni corrections. As mentioned in the abstract 184 CVD associated proteins were measured. Could the authors explain the significant threshold they used for figure 1A and B. 

Dear reviewer, thank you very much for this comment. As we reported in paragraph 2.6 (Statistical analyses), Wilcoxon matched-paired signed rank test was used for intragroup comparison. The significant level was set to p < 0.01. After Bonferroni correction remain significant only LDLR, OPN, PCSK9, MMP-3 and COL1A1 (all P<0.0003) in LLD+/AF+ patients. We replenish this fact to Results.

  1. vi) Figure 1A and B are quite difficult to read with a lot of information. If the authors think it is relevant I would suggest to scaled their data (normalized by sex, age, BMI etc) and present it as forest plot.

Dear reviewer, thank you very much for this comment. Figures 1A and 1B show the intragroup differences between significantly deregulated plasma proteins before and after treatment. Scatter plot, group means (before and after treatment) are indicated by horizontal bars, error bars indicate 95% CI. Figure 1A and 1B in TIFF format with high resolution for better visualization were added to Supplementary material. Data are presented also in Supplementary material Table S1

vii) I would appreciate to read the limitations of the present study (number of patients, T2Diabetes). I am also quite worried about the fact that FH patients above 50 years of age (LLD+AF-) just start lipid lowering drugs at this age. I would like to see this observation discussed as well.

Dear reviewer, thank you very much for this comment. Our study had a relatively small sample size, especially in the lipoprotein aphereses treatment group, and we did not incorporate a control treatment arm in this study. The small number of participants in the present study may affect the accuracy of our results. Thus, our findings need to be verified by multicenter prospective studies. In addition, we cannot exclude the influence of concomitant treatment besides lipoprotein apheresis (statins, ezetimibe, PCSK9 inhibitors) on the measured parameters. Furthermore, since the lipoprotein apheresis treatment technique is carried out only in a few big medical centers, many FH patients are unable to receive apheresis treatment, resulting in a particular bias in patient selection.

In the FH patients under lipid lowering drugs therapy-only the therapy was not started in the later age (55.4±13.1 years), but the patients had wash-out period of four weeks free from their lipid lowering drugs, and the measurements were done after another four weeks the LLD was restarted.

We replenished the text with limitations of our study.

Typos   i) Gene names should be written in italic letters. ii) STAP1 should be removed from the FH causative genes list [see ‘Introduction’ section] (Loaiza et al. 2020, PMID: 31996024) iii) Page 9 line 23: ‘patters’should read as patterns

i/ Thank you very much for this comment. All gene names are written in italic letters now. Protein names are written in standard font.

ii/ Thank you very much for this comment, the sentence was edited. Failure of cosegregation between a rare STAP1 missense variant and hypercholesterolemia has been recently shown not only in the experimental model, but also in the human (Mohebi R, Chen Q, Hegele RA, Rosenson RS. Failure of cosegregation between a rare STAP1 missense variant and hypercholesterolemia. J Clin Lipidol. 2020 Sep-Oct;14(5):636-638. doi: 10.1016/j.jacl.2020.07.010. Epub 2020 Jul 24. PMID: 32828708).

iii/ Typos were corrected.

Reviewer 2 Report

Dlouha D. et al. submitted a study that employed a proteomics approach to examine plasma protein in two patient groups: FH treated with hypolipidemic therapy and FH treated with a combination of therapies. They concluded that hypolipidemic treatment mostly affected endothelium maintenance biomarkers, whereas combination therapy changed proteins involved in cholesterol metabolism and inflammation. This is a very well-written original manuscript, and I only had a few questions to help strengthen it.

  1. There are significant disparities in the length of hypolipidemic treatment between the two groups (table 1). Could this be the huge obstacle and confounding factor?
  2. What antithrombotic medication was provided to the patients? Antithrombotic drugs may have an impact on systemic inflammation.
  3. Was the systemic inflammatory cytokine being measured?
  4. A "test set" and a "validation set" should be used to claim these are "biomarkers." Can these biomarkers predict the prognosis of FH? If the claim of biomarker is based solely on the observation of protein changes, the claims should be soften.

Author Response

Reviewer 2

1.There are significant disparities in the length of hypolipidemic treatment between the two groups (table 1). Could this be the huge obstacle and confounding factor?

Dear reviewer, thank you very much for this comment. Our study aimed to track the intragroup longitudinal changes in plasma protein levels. We did not compare groups LLD+/AF+ and LLD+/AF-. The table 1 is only basic characteristic for both analyzed groups.

  1. What antithrombotic medication was provided to the patients? Antithrombotic drugs may have an impact on systemic inflammation.

Dear reviewer, thank you very much for this comment. With one exception (Godasal 100mg) all subjects have Anopyrin 100mg in prescription.

  1. Was the systemic inflammatory cytokine being measured?

Dear reviewer, thank you very much for this comment. Biomarkers of systemic inflammation and endothelial dysfunction were measured and published in our other papers (for example Víšek J, Bláha M, Bláha V, Lášticová M, Lánska M, Andrýs C, Tebbens JD, Igreja E Sá IC, Tripská K, Vicen M, Najmanová I, Nachtigal P. Monitoring of up to 15 years effects of lipoprotein apheresis on lipids, biomarkers of inflammation, and soluble endoglin in familial hypercholesterolemia patients. Orphanet J Rare Dis. 2021 Feb 27;16(1):110. doi: 10.1186/s13023-021-01749-w. PMID: 33640001; PMCID: PMC7913462), and were reduced for soluble endoglin, hsCRP, and MCP-1, and sP-selectin after each lipoprotein aphereses procedure in some HeFH and HoFH patients.

  1. A "test set" and a "validation set" should be used to claim these are "biomarkers." Can these biomarkers predict the prognosis of FH? If the claim of biomarker is based solely on the observation of protein changes, the claims should be soften.

Dear reviewer, thank you very much for this comment. As we described in "Material and Methods" in the paragraph 2.5, analyses were performed using a high-throughput technique using Olink Proseek® Multiplex CVD II and CVD III panels (https://www.olink.com/products/). Each panel included 92 preselected and validated CVD biomarkers. These panels contain known human cardiovascular and inflammatory markers as well as some exploratory human proteins with great potential as new CVD markers, which were carefully selected in collaboration with leading experts in the field. Each analyte in the panel has been assessed in terms of sample material, specificity, precision, sensitivity, dynamic range, matrix effects and interference.

The biomarker pages (https://www.olink.com/content/uploads/2019/12/Olink-CVD-II-Validation-Data-v2.1.pdf and https://www.olink.com/content/uploads/2019/12/Olink-CVD-III-Validation-Data-v2.1.pdf) include calibrator curves that show the performance of each assay with the estimated sensitivity and dynamic range parameters indicated. These curves are generated during the assay validation process using recombinant antigens, with data presented as Normalized Protein eXpression (NPX) values plotted against protein concentration (in pg/mL).

We replenished this explanation into the paragraph 2.5.

Round 2

Reviewer 2 Report

My entire list of concerns has been handled. Congratulations to all of the authors for their outstanding contributions.